# Effects of Diaphragmatic Breathing on Health: A Narrative Review

**DOI:** 10.3390/medicines7100065

**Published:** 2020-10-15

**Authors:** Hidetaka Hamasaki

**Affiliations:** Hamasaki Clinic, 2-21-4 Nishida, Kagoshima, Kagoshima 890-0046, Japan; h-hamasaki@umin.ac.jp; Tel.: +81-099-2503535; Fax.: +81-099-250-1470

**Keywords:** diaphragmatic breathing, abdominal breathing, breathing exercise, systematic review, randomized controlled trial, respiratory function

## Abstract

**Background:** Breathing is an essential part of life. Diaphragmatic breathing (DB) is slow and deep breathing that affects the brain and the cardiovascular, respiratory, and gastrointestinal systems through the modulation of autonomic nervous functions. However, the effects of DB on human health need to be further investigated. **Methods:** The author conducted a PubMed search regarding the current evidence of the effect of DB on health. **Results:** This review consists of a total of 10 systematic reviews and 15 randomized controlled trials (RCTs). DB appears to be effective for improving the exercise capacity and respiratory function in patients with chronic obstructive pulmonary disease (COPD). Although the effect of DB on the quality of life (QoL) of patients with asthma needs to be investigated, it may also help in reducing stress; treating eating disorders, chronic functional constipation, hypertension, migraine, and anxiety; and improving the QoL of patients with cancer and gastroesophageal reflux disease (GERD) and the cardiorespiratory fitness of patients with heart failure. **Conclusions:** Based on this narrative review, the exact usefulness of DB in clinical practice is unclear due to the poor quality of studies. However, it may be a feasible and practical treatment method for various disorders.

## 1. Introduction

Breathing is an essential part of life. The diaphragm is one of the major respiratory muscles, and its function is vital for proper respiration. At the end of 19th century, Sewall and Pollard [1] firstly investigated the relationship between the movement of diaphragm and chest during respiration. The diaphragm also contributes to vocalization and swallowing, as well as respiration. Its dysfunction is associated with various disorders, such as respiratory insufficiency, exercise intolerance, sleep disturbance, and potential mortality [2,3]. The diaphragm has multiple physiological roles. The phrenic nerve that innervates the functions of the diaphragm has a connection with the vagus nerve, which can affect the whole body system [4]. Diaphragmatic motion in breathing directly and indirectly affects the sympathetic and parasympathetic nervous systems and also influences motor nerve activities and brain mass [5]. The diaphragm also controls the postural stability, defecation, micturition, and parturition by modulating intra-abdominal pressure. Furthermore, its function is associated with metabolic balance [6] and cardiovascular and intraperitoneal lymphatic systems [3].

As diaphragmatic (abdominal) breathing (DB) is a slow and deep breathing method, it should not be considered as just a breathing control [7]. Since time out of mind, traditional martial arts such as tai chi and yoga utilize DB in their practice. DB is defined as breathing in slowly and deeply through the nose using the diaphragm with a minimum movement of the chest in a supine position with one hand placed on the chest and the other on the belly [8]. During breathing, practitioners should be careful that chest remains as still as possible and stomach moves against the hand focusing on contracting the diaphragm. Generally, DB practitioners inhale and exhale for approximately six seconds, respectively. DB is a fundamental procedure during meditation practices in individuals who engage in yoga and traditional martial arts such as tai chi. Recently, a systematic review has reported that mind–body exercise (yoga/tai chi) can reduce stress in individuals under high stress or negative emotions by modulating the sympathetic–vagal balance [9]. Martarelli et al. [10] showed that DB increased the antioxidant activity and reduced the oxidative stress after exercise in athletes. DB has a potential to be a non-pharmacological treatment for patients with stress disorder as well as chronic respiratory disease. Although a number of studies have investigated the efficacy of breathing exercises in treating chronic obstructive pulmonary disease (COPD) [11,12,13,14,15,16,17,18,19,20,21,22,23,24,25,26,27,28,29], asthma [30,31,32,33,34,35], postoperative pulmonary function [36,37,38,39,40], and cardiorespiratory performance in post-Fontan patients [41], the effect of DB on other disorders, for example, cancer, heart failure, and anxiety, still needs to be further investigated. As a martial arts practitioner, the author uses DB in daily mind–body exercises (Figure 1) and feels the necessity to assess whether DB has a favorable impact on the overall health. This review aims to summarize the current evidence of the impact of DB on diseases as described above as well as respiratory function and to discuss its future perspective.

## 2. Methods 

This is a narrative review searching the current evidence on the effect of DB on human health. The author searched the literature on DB using PubMed and Cochrane Library from its inception to May 2020. The search terms (MeSH) were “diaphragmatic,” “breathing exercise,” “systematic review,” and “randomized controlled trial (RCT).” First, the author conducted a search in the systematic reviews, which yielded 19 published articles. Second, the author searched in the RCTs, and this yielded 98 articles. Crossover trials and RCTs already assessed in previous systematic reviews were excluded from this review. The titles and abstracts of the identified articles were reviewed to determine their relevance. Overall, a total of 10 systematic reviews and 15 RCTs were included.

## 3. Results

### 3.1. Systematic Reviews

COPD is the most well-studied disease on which DB has a significant effect. In 2012, the Cochrane Airway Group reported the efficacy of breathing exercises in treating COPD [42]. In this study, 16 RCTs involving 1233 subjects were included with a mean age of 51–73 years and mean forced expiratory volume in 1 s (FEV_1_) of 30–51%, which suggested that the study subjects had moderate-to-severe COPD. Of these, 13 studies were included in the meta-analysis. Primary outcomes were dyspnea, quality of life (QoL), and exercise capacity. Breathing exercises, such as yoga with pranayama timed breathing, pursed-lip breathing, and DB, effectively improved the six-minute walk distance. However, no effects on dyspnea and QoL were observed. Although only two studies [25,43] were included in this systematic review, the four-week supervised DB training improved the six-minute walk distance (mean difference (MD), 34.7 m; 95% confidence interval (CI), 4.1–65.3) [25]. On the other hand, another study reported that DB had an unfavorable effect on dyspnea [43]. Recently, Ubolnuar et al. [44] have also assessed 19 RCTs investigating the efficacy of breathing exercises in patients with any severity stage of COPD. The types of breathing exercise include DB, pursed-lip breathing, just relaxation and slow breathing, ventilatory feedback training, and singing. Overall, the breathing exercises improved the respiratory function such as respiratory rate (RR), tidal volume (V_T_), respiratory time, and QoL of COPD patients. In particular, DB significantly improved the RR (MD, –1.09; 95% CI, −2.19 to 0.00), although the quality of evidence is low [14,29]. However, the QoL measured using the St. George’s Respiratory Questionnaire and dyspnea did not differ between the DB and control groups. Furthermore, these results indicate that, although breathing exercises including DB are promising to improve the exercise capacity and respiratory function, their effects on clinical symptoms and QoL are inconsistent due to the severity stage of COPD. 

The Cochrane Airway Group reevaluated the efficacy of breathing exercises in adults with asthma in 2020 [45]. Nine studies were added to the previous systematic review published in 2013, and a total of 22 RCTs were included in this systematic review and meta-analysis. Unfortunately, since only one RCT met the inclusion criteria [46], the effect of DB on QoL and asthma symptoms was inconclusive. However, breathing exercises such as yogic breathing and the Buteyko breathing technique had positive effects on QoL and asthma symptoms. Moreover, breathing exercises improved the QoL measured using the Asthma Quality of Life Questionnaire at three months (MD, 0.42; 95% CI, 0.17–0.68) and at six months (odds ratio, 1.34; 95% CI, 0.17–0.68) compared with no active control. Furthermore, hyperventilation symptoms and FEV_1.0%_ were predicted to be improved by breathing exercises.

Prem et al. [47] investigated the effect of DB on the QoL of patients with asthma. Only three RCTs assessing the effect of DB on asthma were included [30,31,32]. The intervention used in the study by Thomas et al. [31] was DB plus nasal breathing exercise. In addition, the interventions as controls were asthma education [30,31] and conventional asthma medication [32]. This systematic review did not perform a meta-analysis. However, DB improved the QoL measured using the Asthma Quality of Life Questionnaire; specifically, the questionnaire score was improved: 0.79 [30] and 1.12 [31]. Moreover, the scores of the Asthma Control Test (from 18 ± 2.5 to 22 ± 3.3) and end-tidal CO_2_ (by 4 mmHg) were improved in the study by Grammatopoulou et al. [32]. The authors suggested that DB improved the QoL of patients with asthma based on the reduction in hyperventilation, which physiologically improved the respiratory function. 

The effect of breathing exercise in children with asthma was systematically reviewed in 2016 [48]. Only three studies [33,34,35] were eligible for this systematic review. The primary outcomes were QoL, asthma symptoms, and adverse events. None of these studies evaluated the single effect of DB, and the breathing exercise programs consisted of DB, lateral costal breathing [33], pursed-lip breathing [35], and endurance exercise [34]. A heterogeneity in the asthma severity of patients among the studies was observed. The difference in the primary outcomes could not be found in the comparisons between the intervention and control groups. Lima et al. [35] reported that the peak expiratory flow (PEF) was improved after the intervention, but based on the meta-analysis, no clear evidence could confirm that DB improved the respiratory function. Moreover, it was inconclusive that DB had a benefit or risk in children with asthma. 

Dysfunctional breathing is associated with poor asthma control in children [49,50]. Barker et al. [51] assessed the effect of breathing exercises in children with dysfunctional/hyperventilation syndrome. However, no eligible studies were found for this systematic review. This lack of evidence is due to the insufficient number of well-designed RCTs conducted in children. On the other hand, Jones et al. [52] evaluated the effect of breathing exercises in adults with dysfunctional/hyperventilation syndrome. Since only a single RCT [53] met the inclusion criteria, this systematic review could not provide a reliable conclusion regarding the effect of DB on dysfunctional breathing. The included study enrolled 45 patients with hyperventilation syndrome and divided them into three groups (relaxation therapy, relaxation therapy and DB, and control) of 15 patients each. As the frequency and severity of hyperventilation attacks were significantly reduced in the DB group compared with the control group, no detailed data and statistical analysis were presented in this study [53]. Therefore, the effect of DB on dysfunctional breathing is still unclear. 

A systematic review with meta-analysis examined (1) the generalizability, consistency, volume, and quality of the evidence for breathing control; and (2) the effect of breathing control on various clinical outcomes [54]. This systematic review included a total of 20 studies: 2 RCTs [55,56], 3 non-RCTs [57,58,59], and 15 quasi-experimental studies [14,60,61,62,63,64,65,66,67,68,69,70,71,72,73]. The study participants were also heterogeneous; 80% of the studies recruited patients with chronic respiratory disease, such as COPD, and 20% of the studies included patients with other conditions (e.g., post-surgery, chronic progressive multiple sclerosis) and asymptomatic individuals. DB was required to be the single intervention used in all studies. DB had beneficial effects on abdominal excursion (MD, 1.36; 95% CI, 0.42–2.31), diaphragm excursion (MD, 1.39; 95% CI, 1.00–1.77), short-term changes in respiratory function, RR (MD, −0.84; 95% CI, −1.09 to 0.60), V_T_ (MD, 0.98; 95% CI, 0.71–1.25), gas exchange, arterial oxygen saturation (MD, 0.63; 95% CI, 0.25–1.02), and percutaneous oxygen (MD, 1.48; 95% CI, 0.85–2.11). On the other hand, DB had a negative impact on the work of breathing (MD, 1.06; 95% CI, 0.52–1.60) and dyspnea (MD, 1.47; 95% CI, 0.88–2.05) in patients with severe respiratory disease. DB had no significant effects on ventilation, long-term change in respiratory function, vital capacity (VC), forced vital capacity (FVC), expiratory flow rate, FEV_1_, respiratory muscle strength, oxygen consumption, respiratory muscle efficiency, ventilation distribution, and 12-min walk test. On the other hand, DB was effective in the short-term improvement of respiratory function, but it did not have a beneficial effect on the long-term physiological outcomes and energy cost of breathing. Interestingly, DB could negatively affect the respiratory symptoms of patients with severe respiratory disease and may not be applicable to all kinds of respiratory disease. However, the generalizability and quality of evidence is not high as this systematic review included only two RCTs and the heterogeneity of the characteristics of study subjects and the intervention methods used among the studies was large.

Grams et al. [74] examined the effects of breathing exercises on the prevention of postoperative pulmonary complications and recovery of pulmonary function in patients who had upper abdominal surgery. A total of six RCTs or quasi-RCTs were included in this systematic review [36,37,38,39,40,75], four of which were conducted in Brazil. The meta-analysis showed that the maximal expiratory pressure and maximal inspiratory pressure increased by 12.8 (95% CI, 7.6–18.1) and 5.6 (95% CI, 0.6–10.5) mmH_2_O, respectively, on Day 1 postop [38,39,40]. However, DB was observed to have no significant effects on respiratory function including FVC, FEV, and FEV_1_. This systematic review indicates that breathing exercises, which mainly consist of DB, improve the respiratory muscle strength of patients after upper abdominal surgery. However, the included studies investigated the effect of DB on Day 1–5 postop, and the respiratory functions of the study subjects at baseline were heterogeneous. Therefore, the findings of this systematic review are limited to a specific circumstance and the generalizability is low. 

Recently, Hopper et al. [76] reported that DB might have reduced the physiological and psychological stress, although the meta-analysis could not be performed due to the methodological heterogeneity and outcome measures. One RCT [77] and quasi-experimental studies [78,79] were included in this qualitative analysis. Ma et al. [77] reported that DB reduced the RR and salivary cortisol levels in an RCT, suggesting that DB has a favorable effect on stress. Two experimental studies also showed that DB was effective for improving the blood pressure control [78] and stress measured using the Depression Anxiety Stress Scale-21 [79]. However, more well-designed RCTs with an appropriate sample size are needed to conclude whether DB is beneficial for reducing stress. 

Table 1 summarizes the results of these systematic reviews. 

### 3.2. Randomized Controlled Trials

#### 3.2.1. COPD and Asthma

It is apparent that previous studies investigating the effects of DB have been conducted in patients with COPD. The author has identified a recent RCT that was not included in previous systematic reviews. Yekefallah et al. [80] compared the effect of the breathing exercise involving DB and pursed-lip breathing and upper limb exercise on exercise capacity measured through a six-minute walking test in patients with COPD. Seventy-five patients with moderate-to-severe COPD were recruited and divided into three groups: upper limb exercise group (*n* = 25), breathing exercise group (*n =* 25), and control group (*n* = 25). Participants in the breathing exercise group performed DB and pursed-lip breathing for one minute, respectively, with a one-minute rest between these exercises. They were asked to do these exercises four times a day for four weeks. On the other hand, participants in the upper limb exercise group performed upper limb strengthening exercises using dumbbells for 20 min per session, thrice a week, for four weeks. Moreover, all participants completed the study. The mean walking distance significantly increased in the breathing exercise group (from 355.3 ± 47.9 m to 376.9 ± 37 m) and in the upper limb exercise group (from 389.8 ± 5.8 m to 409.5 ± 29.8 m) during the study, whereas the control group did not show any significant change. Although both the upper limb exercise and DB plus pursed-lip breathing were effective in increasing the walking distance, a post hoc analysis revealed that the walking distance of the upper limb exercise group was longer than that of the breathing exercise group. This study indicates that the upper limb strengthening exercise is more effective for improving the exercise capacity of COPD patients than DB training.

The respiratory function and abdominal and thoracic kinematics changes due to DB training in patients with moderate persistent asthma were evaluated, although the intervention might be a respiratory muscle training rather than a simple DB training [81]. Eighty-eight inactive patients with asthma aged between 18 and 34 were enrolled in this RCT. The study participants were categorized into aerobic exercise (*n* = 22), DB (*n* = 22), aerobic exercise combined with DB (*n* = 22), and control (*n* = 22) groups. The participants in the intervention groups performed the training program thrice a week for eight weeks. The DB training in this study was unique. The participants in the DB group breathed using a tube to maximize their inspiration and expiration, and a 2.5 kg weight (Week 1–4) or a 5 kg weight (Week 5–8) was put on their abdominal cavity. Moreover, they completed three sets of 5–10 repetitions using one second of inspiration and two seconds of expiration, three sets of 10–15 repetitions using two seconds of inspiration and four seconds of expiration, and three sets of 15–20 repetitions using three seconds of inspiration and six seconds of inspiration. The participants in the aerobic exercise group walked and/or jogged for 30 min at the intensity of 60% of the age-predicted maximum heart rate. After the eight-week intervention, the DB training improved the FVC (from 3.01 ± 0.58 L to 3.52 ± 0.74 L), FEV_1_ (from 2.85 ± 0.57 L to 3.22 ± 0.63 L), FEV_1_/FVC ratio (from 94.86 ± 4.94% to 90.64 ± 6.67%), PEF (from 7.10 ± 1.57 L to 7.68 ± 1.26 L), and inspiratory VC, but the forced expiratory flow (FEF) rate, maximum voluntary ventilation (MVV), and V_T_ did not change. On the other hand, aerobic exercise improved the FVC (from 2.77 ± 0.48 to 3.11 ± 0.71 L), FEV_1_ (from 2.72 ± 0.53 to 2.97 ± 0.65 L), PEF (from 7.15 ± 1.45 L to 7.57 ± 1.47 L), MVV (from 103.65 ± 27.86 L/min to 128.97 ± 27.56 L/min), and inspiratory VC, but the FEV_1_/FVC ratio, FEF, and V_T_ did not change. Aerobic exercise combined with DB more effectively improved the FVC (from 2.87 ± 0.67 L to 3.68 ± 0.82 L) and FEV_1_ (from 2.70 ± 0.67 L to 3.30 ± 0.70 L) than aerobic exercise alone, but DB and aerobic exercise were equally effective in the improvement of FVC and FEV_1_. Aerobic exercise, DB, and DB combined with aerobic exercise improved the chest circumferences during inspiration, but no significant improvement was observed during the rest and expiration phases. Interestingly, DB improved the resting, inspiratory, and expiratory abdominal circumferences at the height of the midpoint between the umbilicus and the xiphoid process, but aerobic exercise did not change the resting circumference.

#### 3.2.2. Cancer

Campbell et al. [82] investigated the efficacy of relaxation techniques in treating the eating problems of cancer patients who have a prognosis of at least six months and have nutritional problems such as weight loss. The relaxation technique includes DB, autosuggestion, relaxing of muscles, and image control. The changes in weight and performance status measured using the Karnofsky Performance Status Scale during the study period were assessed. Twenty-two patients with cancer were randomly assigned to the intervention (*n* = 12) and control groups (*n* = 10), respectively. After performing the relaxation training for six weeks, 75% of the patients gained weight within 10% of one’s ideal weight. Performance status was improved in 33% of the patients after the eight-week intervention. Moreover, relaxation training using DB may support the treatment of eating problems in patients with cancer. 

Shahirai et al. [83] evaluated the effect of DB, muscle relaxation, and body image on the QoL of older patients with breast or prostate cancer. Fifty patients were recruited and categorized into the intervention (*n* = 25) and control (*n* = 25) groups. The functional QoL score was immediately improved after the intervention (from 31.6 to 60.5 points) and six weeks after the intervention (from 31.6 to 66 points), whereas no significant changes were observed in the control group. Furthermore, the mean score of the general domain of QoL was also immediately increased after the intervention (from 36.33 to 64.33 points) and six weeks after the intervention (from 36.33 to 52.33 points). On the other hand, it was decreased in the control group during the study period. These studies applied the use of concurrent techniques, but not DB techniques, and the outcome measures were mortality and survival period. Thus, whether DB is useful for cancer treatment or not is inconclusive. However, relaxation and DB techniques may be a cost-effective and convenient method for improving the general condition of patients with cancer.

#### 3.2.3. Other Diseases

Silva and Motta [84] investigated the effect of DB, abdominal muscle training, and massage on pediatric patients with chronic functional constipation. Seventy-two patients aged 4–18 were categorized into the physiotherapy plus medication (*n* = 36) and the medication using only laxatives (*n* = 36) groups. The physiotherapy consisted of DB, isometric training of the abdominal muscles to increase intra-abdominal pressure, and slow circular clockwise abdominal massage. After the six-week intervention, the defecation frequency was significantly higher in the physiotherapy group than the medication-only group. Furthermore, DB may increase intra-abdominal pressure and stimulate the parasympathetic activity, which increases the colonic motility and improves the defecation frequency. 

Wang et al. [85] investigated the effect of DB on blood pressure in prehypertensive patients. Twenty-six postmenopausal women aged 45–60 were enrolled and categorized into the intervention and control groups. Twenty-two participants (intervention group, n =12; control group, *n* = 10) completed the study. The intervention group was treated with DB combined with the frontal electromyographic biofeedback-assisted relaxation training, whereas the control group only performed DB techniques. All participants performed 10 sessions of treatment once every 3 days. After the training, in the intervention group, systolic and diastolic blood pressures were decreased by 8.4 and 3.9 mmHg, respectively. Single DB also significantly decreased the systolic blood pressure by 4.3 mmHg, but no changes in the diastolic blood pressure were observed. DB combined with the biofeedback training was more effective in lowering the blood pressure than DB alone. In addition, the RR interval increased during the training in the intervention group, whereas no change was observed in the control group. The standard deviation of the normal–normal intervals significantly increased in both groups. Although DB alone was effective in lowering the blood pressure and improving the heart rate variability, the biofeedback training seemed to strengthen its effect through inhibiting sympathetic activity and improving vagal tone [86]. 

Seo and colleagues [87] examined the effect of DB on dyspnea and physical activity of patients with heart failure. Thirty-six patients were enrolled in this study and were categorized into the home-based DB retraining (*n* = 18) and control (*n* =18) groups. A total of 29 patients (intervention group, *n* = 13; control group, *n* = 16) completed the study, and 27 patients (intervention group, *n* = 12; control group, *n* = 15) who continued the home-based DB retraining were followed up for five months. After the eight-week intervention, the DB group showed little improvement in dyspnea. The functional status in the DB and control groups increased by 10.5% and 4.4%, respectively, but declined by 2.2% in the control group in the five-month follow-up. On the other hand, the average daily activity measured by a triaxial accelerometer, ActiGraph, significantly increased by 14% in the DB group and decreased by 6% in the control group. No adverse effects were reported. Moreover, DB was a feasible treatment option for patients with heart failure. Daily physical activity can be increased due to the improvement of dyspnea through regular DB exercise, which may lead to maintaining or improving the cardiorespiratory fitness of patients with heart failure. Furthermore, the results of the studies by Wang et al. [85] and Seo et al. [87] indicate that DB has beneficial effects on cardiovascular health. 

Sutbeyaz and colleagues [88] conducted an interesting RCT that compared the efficacy of DB and pursed-lip breathing in inspiratory muscle training for improving the cardiopulmonary functions of patients with subacute stroke. Forty-five inpatients with stroke were recruited and categorized into the breathing retraining (*n* = 15), inspiratory muscle training (*n* = 15), and control (*n* = 15) groups. The breathing training program consisted of 15 min of DB combined with pursed-lip breathing, 5 min of air-shifting techniques, and 10 min of voluntary isocapnic hyperpnea. The participants received daily training, six times a week, for six weeks. No significant changes in VC, FVC, FEV_1_, FEF_25–75%_, and MVV from baseline in the DB group were observed, but the PEF of the DB intervention group improved as compared with both the inspiratory training and control groups. In contrast, inspiratory muscle training significantly improved VC, FVC, FEV_1_, FEF_25–75%_, and MVV as compared with controls. Interestingly, DB increased both the maximum inspiratory and expiratory pressures, but inspiratory muscle training did not increase the maximum expiratory pressure. In contrast to DB, inspiratory muscle training improved the exertional dyspnea and functional status based on the Barthel Index and Functional Ambulation Category scores. The general health, pain, vitality, and emotional role domains of the SF–36 improved in the DB group from baseline as compared with the control group. The short-term inspiratory muscle training effectively improved the respiratory function and exercise capacity of patients with stroke, but DB was also effective in improving the PEF, inspiratory and expiratory pressures, and QoL. Considering that inspiratory muscle training requires the appropriate medical equipment, DB is a more feasible treatment option for improving the cardiopulmonary function of inpatients with stroke. 

Eherer and colleagues [89] assessed the effect of the four-week DB training on the QoL, pH-metry, and on-demand proton pump inhibitor usage of patients with nonerosive gastroesophageal reflux disease (GERD). Nineteen patients were enrolled in this RCT and were categorized into the training (*n* = 10) and control (*n* = 9) groups. The training group engaged in daily DB practice for at least 30 min. After the four-week DB training, the time with a pH < 4.0 in the training group decreased from 9.1% ± 1.3% to 4.7% ± 0.9%, and the QoL scores measured using the GERD Health-Related Quality of Life Scale also improved from 13.4 ± 1.98 to 10.8 ± 1.86, but no changes in the control group were observed. Furthermore, after the nine-month follow-up, patients who continued the DB techniques showed an improvement in their QoL scores (from 15.2 ± 2.2 to 9.7 ± 1.6) and proton pump inhibitor usage (from 98 ± 34 mg/week to 25 ± 12 mg/week). Furthermore, DB as a non-pharmacological intervention was observed to reduce the proton pump inhibitor usage and improve the long-term QoL of patients with GERD. 

In 2005, an interesting RCT was conducted in India [90]. Migraine is a common but hard-to-treat disease. Kaushik et al. investigated whether biofeedback-assisted DB could treat migraine. This study enrolled 192 patients who were then categorized into biofeedback (*n* = 96) and control (*n* = 96) groups. Moreover, 24 (25%) patients in the biofeedback group were excluded. The control group received 80 mg/day of propranolol, whereas the biofeedback group was subjected to DB and relaxation with electromyogram and temperature feedback for six months. Biofeedback-assisted DB was effective in 66.66% of the patients. In both groups, the severity, frequency, number of vomiting episodes, and duration of attacks were decreased. One year after the intervention, the resurgence of migraine was observed in 9.37% of the participants in the biofeedback group, which was significantly lower than that of the propranolol group (38.54%). Differences in the resurgence rate between the groups were observed (*p* < 0.001). In the propranolol group, adverse effects such as fatigue and nausea were observed in 13.54% of the patients, whereas the side effects were only observed in 5.2% of the patients in the biofeedback group. The authors recommended that biofeedback-assisted DB and relaxation techniques should be used as a treatment for migraine. 

A systematic review has shown that DB may be useful for stress management [76]. Chen et al. [91] evaluated the effectiveness of DB training program on anxiety. Anxiety is associated with respiratory symptoms such as dyspnea, shallow respiratory breathing, hyperventilation, and chest tightness [92], as well as cardiovascular symptoms such as tachycardia and palpitations [92,93]. The authors hypothesized that relaxation and DB techniques could reduce anxiety. Forty-six individuals who had anxiety for at least a month were recruited, but only 30 participants (DB group, *n* = 15; control group, *n* = 15) completed the eight-week study. The DB group practiced DB at least twice a day and 10 exercises per session. The anxiety scores measured using the Beck Anxiety Inventory declined from baseline (19.13 ± 7.52) to week 4 (12.67 ± 7.09) and also from week 4 to week 8 (5.33 ± 4.52). Moreover, after the eight-week DB training, the peripheral temperature increased from 33.26℃ ± 1.49℃ to 34.77℃ ± 1.01℃, heart rate decreased from 85.52 ± 8.0 to 72.45 ± 5.57 beats/min, and breathing rate decreased from 16.24 ± 2.27 to 12.59 ± 2.40 breaths/min, whereas no significant changes in the control group were observed. Furthermore, DB is effective to reduce anxiety, which leads to favorable changes in physiological indicators.

Table 2 summarizes the results of these RCTs.

#### 3.2.4. Healthy Individuals

The effects of DB on healthy individuals have been also investigated. An experimental study was conducted to investigate whether DB had an impact on motion sickness in a virtual reality environment [94]. Healthy individuals were screened for motion sickness susceptibility. A total of 60 motion sickness susceptible subjects were randomly categorized into the DB (*n* = 31) and control (*n* = 29) groups. The participants wore 3D goggles and experienced motion sickness in a virtual reality space (10-min fluctuating view of a stormy sea). During the virtual reality experience, the respiration rate was significantly lower (11.38 ± 3.49 breaths/min vs. 16.21 ± 2.77 breaths/min) and the heart rate variability (respiratory sinus arrhythmia) was significantly higher (7.46% ± 1.05% vs. 6.38% ± 0.86%) in the DB group compared with the control group. In addition, the self-reported motion sickness rating (1.37 ± 0.44 vs. 1.78 ± 0.63) and the motion sickness assessment questionnaire score (2.1 ± 0.91 vs. 2.85 ± 1.72) were significantly lower in the DB group than those in the control group. In the DB group, a positive correlation between the respiration rate and motion sickness rating and negative relationships of the heart rate variability with respiration rate and motion sickness rating were observed. Therefore, these findings suggested that DB increased the parasympathetic nervous system activity, decreased the respiration rate, and improved the motion sickness symptoms.

Gimenez et al. [95] compared the effectiveness of comprehensive directed breathing retraining with DB on male smokers who had exertional dyspnea but normal spirometry. Twenty-four active male smokers aged 33–60 were enrolled and categorized into the experimental (comprehensive directed breathing retraining) and control (DB) groups. Both groups performed 60 min of DB, 30 min of walking, and conditioning exercises for 5 days a week for 4 weeks. The participants were asked to continue DB at home, walking, and exercises twice daily for at most 30 min. The experimental group was educated about the anatomy and physiology of respiration, enhanced their awareness on abnormal breathing patterns, shown the ventilator rhythm on a spirogram, and watched a DB instructional film. The measurement of physiologic parameters was performed at rest and at 40-W exercise for 10 min. In the experimental group, 34 of 44 lung function parameters, such as dyspnea index, ventilation capacity, FEV_1_, PEF, VO_2_, VCO_2rest_, and PaO_2_, were improved. The single DB intervention did not effectively improve the exertional dyspnea and lung function. Moreover, the authors referred to an unfavorable effect of DB that previous studies on patients with COPD had shown: the possibility that DB worsened the chest wall motion, reduced the efficiency of ventilation, and increased the respiratory workload [64,96,97]. 

Han and Kim [98] investigated the effect of DB combined with upper extremity exercise on the lung function of young healthy individuals. Forty male adults were recruited and categorized into the experimental (DB with upper extremity exercise; *n* = 20) and control (only DB; *n* = 20) groups. Both groups performed 10 min of warm-up exercise, 5 min of DB, and 10 min of cool-down exercise. Additionally, the experimental group performed breathing exercises with 25 min of dynamic upper extremity exercise using an elastic band with 40% resistance for one repetition maximum, whereas the control group performed 25 min of regular breathing exercise. Both groups performed the exercise session thrice a week for four weeks. After the four-week training, FVC significantly increased in both experimental and control groups. FEV_1_ and PEF did not change in both groups. However, FEV_1_ increased by 0.05 L in the experimental group, whereas it decreased by 0.02 L in the control group. This study indicates that DB is effective in improving FVC, but the upper extremity exercise may have an additional effect on obstructive ventilatory disturbance.

Bahensky et al. [99] investigated how DB based on yoga affects the efficiency of breathing in adolescent endurance runners. This study included 37 runners who performed endurance training at least six times a week. The intervention group (*n* = 21) engaged in DB exercise for at least 10 min per session, at least 5 times a week, for 4 months. The V_T_ and breathing frequency were measured at two and four months after the intervention started. A spiroergometry test was performed using a bicycle ergometer at the point of subjective exhaustion. In the intervention group, V_T_ significantly increased from 2.02 ± 0.43 L at baseline to 2.11 ± 0.43 L after the two-month intervention and to 2.25 ± 0.51 L after the four-month intervention, whereas no changes in the control group were observed. Moreover, the breathing frequency significantly increased from 59.0 ± 8.6 breaths/min at baseline to 55.6 ± 9.5 breaths/min after the two-month intervention and to 52.2 ± 9.2 breaths/min after the four-month intervention, whereas no changes in the control group were observed. The DB training for four months effectively increased the V_T_ by 10.96% and decreased the breathing frequency by 11.47%. DB may improve the endurance capacity of the respiratory muscles in healthy adolescents.

Previous studies have shown that DB has no significant impacts on aerobic capacity in healthy individuals. Respiratory muscle trainings also appear to have no beneficial effects on VO_2_max in healthy non-smokers [100] and athletes [101].

## 4. Discussion

DB has various physiological effects in humans. The diaphragm is the major respiratory muscle. As the movement of the diaphragm has a positive correlation with the lung volume [102], using the diaphragm consciously during respiration increases the lung capacity. DB facilitates slow respiration, but if RR decreases, hypercapnia and the activation of chemoreceptors would be induced to increase RR to maintain the respiratory homeostasis [103]. DB that controlled RR at six breaths/min reduces the chemoreflex response to hypoxia and hypercapnia compared with normal breathing [104]. Decreased RR increases the V_T_, which improves the efficiency of ventilation for oxygen [105] through alveolar recruitment and distention, improving the alveolar ventilation due to reduced alveolar dead space and increasing the arterial oxygen saturation [103]. Therefore, DB has a potential to improve the blood oxygen levels.

RR also affects the heart rate, systemic blood pressure, and circulating blood volume. Generally, inspiration decreases the intrathoracic pressure and increases the pressure gap between the right heart and the systemic circulation, which increases the venous return to the right heart. On the other hand, the pulmonary venous return decreases and the blood volume in the left heart is reduced. As a result, the cardiac output increases due to the increase of blood volume in the right heart. This physiological action is reversed in expiration [103]. Heart rate increases during inspiration and decreases during expiration while arterial blood pressure is lowered [106]. DB enhances the fluctuations in blood pressure and heart rate [103] via slow breathing [107] and diaphragm excursions, therefore improving the baroreflex sensitivity, heart rate variability, and blood pressure oscillations [103,108].

Breathing has a close relationship with autonomic nervous system function. The phrenic nerve that controls the movement of the diaphragm is connected to the vagus (parasympathetic) nerve [4]. Decreasing the RR by DB activates the parasympathetic nervous activity while suppressing the sympathetic nervous activity [11]. Chang et al. [109] reported that slow breathing with eight breaths/min makes the balance of the parasympathetic nervous activity dominant. Autonomic dysfunction, for example, a reduction in heart rate variability, is associated with an increased risk of cardiovascular mortality and morbidity [110]. Hyperactive sympathetic nervous activity and hypoactive parasympathetic nervous activity can be regulated by DB, which will improve the cardiovascular health. In addition, yoga practice tends to tune the brain toward a parasympathetically driven mode and positive states [111]. Jerath et al. [112] indicated that breathing stimulated the vagal activation of gamma-aminobutyric acid pathways in the brain, and reduced stress and anxiety. Furthermore, DB appears to have a favorable effect on the cardiovascular system and brain through the improvement of the autonomic balance.

Although the current evidence regarding the effects of DB on human health is accumulating, several limitations should be considered to conclude its efficacy in clinical practice. Firstly, the DB technique among the studies has not been standardized. The inspiratory and expiratory phase times ranged from 4 to 8 s, respectively, and the practitioners performed DB in various postures such as supine position, semi-recumbent position, or seated position. Moreover, the optimal RR and posture for achieving physiological benefits are still unknown. Secondly, the effect of DB may differ depending on the severity of the diseases. For instance, DB could be harmful for dyspnea in patients with severe COPD. Therefore, future studies should investigate whether the effects of DB differ according to the severity of diseases. Thirdly, considerable heterogeneity among studies was observed, such as the characteristics of the study subjects, intervention frequency and duration, and controls. Furthermore, previous systematic reviews assessed in this study have different criteria of inclusion. For example, several studies that did not investigate the single effect of DB were included in a systematic review (e.g. Thomas et al. [31]). Fourthly, systematic reviews included a wide range of studies from the 1950s to 2010s. Studies that were performed 50 years ago have important information; however, recent studies may be more important because statistical methods and quality of data advances with the times. Such heterogeneity might cause the findings of this review to be inconclusive. Finally, the primary outcomes of systematic reviews are usually clinical symptoms, QoL, respiratory function, and exercise capacity, and no studies have evaluated the effect of DB on hard endpoints, such as the development of respiratory failure, cardiovascular disease, and mortality. Most of the studies have short study periods (e.g., 4–6 weeks). Thus, the long-term effect of DB should be clarified in the future. Despite these limitations, DB has the potential to improve various kinds of disease. Moreover, no serious adverse effects have also been reported in the RCTs. Recently, a number of studies have shown that physical rehabilitation improves exercise capacity in transplant recipients and candidates [113,114]. DB could also be safe and feasible in the post-transplant management due to its non-invasive technique.

## 5. Conclusions

Previous systematic reviews and meta-analyses have shown that DB is effective for improving the exercise capacity and RR in patients with COPD. On the other hand, DB could deteriorate dyspnea in severe COPD patients. Moreover, the effect of DB on the QoL of patients with asthma still needs to be investigated further. DB may also be beneficial for reducing both physiological and psychological stress and could improve the respiratory function and respiratory muscle strength, but more firm evidence will be needed in the future. In addition, DB may help in treating eating disorders, chronic functional constipation, hypertension, migraine, and anxiety, as well as the QoL of patients with cancer and GERD and the cardiorespiratory fitness of patients with heart failure. Furthermore, DB could be a feasible and practical technique for patients with such disorders. Although further studies are needed to clarify the effects of DB on human health, DB can support clinical practice.

## Figures and Tables

**Figure 1 medicines-07-00065-f001:**
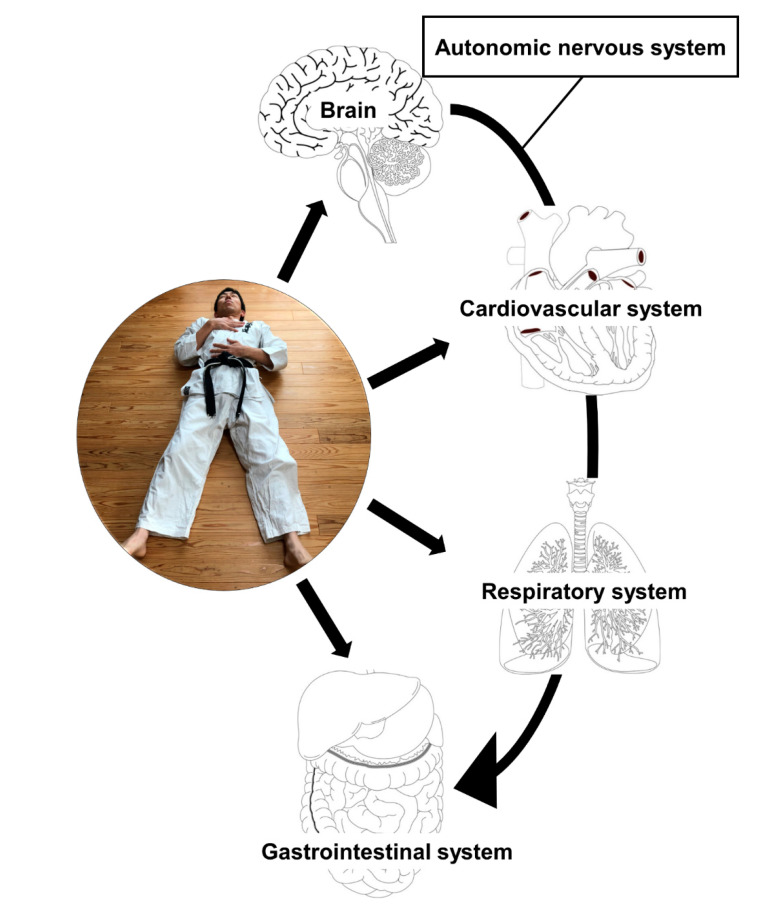
Breathing in slowly and deeply through the nose with a minimum movement of the chest in a supine position with one hand placed on the chest and the other on the belly. Diaphragmatic breathing has an impact on the brain and cardiovascular, respiratory, and gastrointestinal systems through the modulation of the autonomic nervous function.

**Table 1 medicines-07-00065-t001:** Systematic reviews and meta-analyses assessing the effects of diaphragmatic breathing on various disorders.

Authors, Year	Subjects	Included Studies	Primary Outcomes	Results
Holland et al., 2012 [42]	COPD	16 RCTs	Dyspnea, exercise capacity, and health-related quality of life	Dyspnea↑ 6-min walk distance↑
Ubolnuar et al., 2019 [44]	COPD	19 RCTs	Ventilation, exercise capacity, dyspnea, and quality of life	Respiratory rate↓ Quality of life→
Santino et al., 2020 [45]	Asthma	22 RCTs	Quality of life	Unknown due to insufficient data
Prem et al., 2013 [47]	Asthma	3 RCTs	Quality of life	Quality of life↑? (meta-analysis was not performed)
Macêdo et al., 2016 [48]	Asthma (children)	3 RCTs	Quality of life, asthma symptoms, and adverse effects	Unknown due to insufficient data
Barker et al., 2013 [51]	Dysfunctional/hyperventilation syndrome (children)	No eligible studies	Quality of life	Unknown
Jones et al., 2013 [52]	Dysfunctional/hyperventilation syndrome (adults)	1 RCT	Quality of life and adverse effects	Unknown due to insufficient data
Lewis et al., 2007 [54]	Chronic respiratory disease, post-surgical, or asymptomatic individuals	2 RCTs, 3 non-RCTs, and 15 quasi-experimental studies	Short-term clinical outcomes (not specified)	Abdominal excursion↑, diaphragm excursion↑, respiratory rate↓, tidal volume↑, arterial oxygen saturation↑, percutaneous oxygen↑ Work of breathing↑, dyspnea↑
Grams et al., 2012 [74]	Post upper abdominal surgery	6 RCTs or quasi-RCTs	Respiratory function and postoperative complications	Respiratory muscle strength↑ Respiratory function→
Hopper et al., 2019 [76]		1 RCT and 2 quasi-experimental studies	Stress	Stress↓ (meta-analysis was not performed)

COPD, chronic obstructive pulmonary disease; RCT, randomized controlled trial; ↑, increase; →, no change; ↓, decrease.

**Table 2 medicines-07-00065-t002:** Randomized controlled trials assessing the effects of diaphragmatic breathing on various disorders.

Authors, Year	Subjects	Intervention Study Duration	Results
Yekefallah et al., 2019 [80]	75 patients with COPD	Breathing exercise (DB and pursed-lip breathing) and upper limb exercise One month	6-min walking distance↑
Shaw and Shaw, 2011 [81]	88 patients with asthma	DB, aerobic exercise, and aerobic exercise plus DB 8 weeks	FVC↑, FEV_1_↑, FEV_1_/FVC ratio↓, PEF↑, FEF rate→, MVV→, V_T_→
Campbell et al., 1984 [82]	22 patients with cancer (except breast cancer)	Relaxation technique including DB 6 weeks	Desirable weight gain Improvement in Performance Status
Shahirai et al., 2017 [83]	50 elderly patients with breast or prostate cancer	Muscle relaxation, guided imagery, and DB 6weeks	Quality of life↑
Silva and Motta, 2013 [84]	72 pediatric patients with chronic functional constipation	Isometric training of the abdominal muscle, DB, and abdominal massage 6 weeks	Defecation frequency↑ Fecal incontinence→
Wang et al., 2010 [85]	22 postmenopausal women with prehypertention	DB and DB with frontal electromyographic biofeedback training	Systolic blood pressure↓
Seo et al., 2016 [87]	29 patients with heart failure	Home-based DB retraining 8 weeks	Dyspnea↓ Daily physical activity↑ Functional status↑
Sutbeyaz et al., 2010 [88]	45 inpatients with subacute stroke	DB combined with pursed-lip breathing and inspiratory muscle training 6 weeks	PEF ↑, VC→, FVC→, FEV_1_→, FEF_25–75%_→, MVV→
Eherer et al., 2012 [89]	19 patients with non-erosive gastroesophageal reflux disease	DB 4 weeks	Time with a pH < 4.0↓ Quality of life↑
Kaushik et al., 2005 [90]	167 patients with migraine	DB with biofeedback and 80 mg/day of propranolol 6 months	Resurgence of migraine↓
Chen et al., 2017 [91]	30 patients with anxiety	DB 8 weeks	Anxiety score↓, peripheral temperature↑, heart rate↓, breathing rate↓

COPD, chronic obstructive pulmonary disease; DB, diaphragmatic breathing; VC, vital capacity; FVC, forced vital capacity; FEV_1_, forced expiratory volume in 1 s; PEF, peak expiratory flow; FEF, forced expiratory flow; MVV, maximum voluntary ventilation; V_T_, tidal volume; ↑, increase; →, no change; ↓, decrease.

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
