# Peer review of "Effects of Diaphragmatic Breathing on Health: A Narrative Review"

_medicines, 2020, doi:10.3390/medicines7100065_

Round 1

Reviewer 1 Report

The manuscript is interesting  in the  fields  of  the  exercise prescription application,  in Sports  Medicine  and also  in the cardiological  and internal  medicine . It can be improved with the addition  of more details  of  the  potential aplication in the metabolic chronic diseases  where also  the simpato-vagal balance can be affected: as in case of  diabete , metabolic syndrome and obesity. 

In addition it could be helpfull  to give some information about  the impact of  the  model  to the VO2 max parameter, if avaliable  in literature .

 The  same aspect  could be higlighted in a session dedicated to the congenital cardiopaties, especially  in the post surgical treatment .

Also  in the potential  application in the post   surgical  solid organ transplantation managment by the  physical  activity program .

This last  aspect has been  only partially evaluated in a previous study that  you  can insert in the text  just  to  start  with  the discussion. 

Renal function and physical fitness after 12-mo supervised training in kidney transplant recipients Giulio Sergio Roi, Giovanni Mosconi, Valentina Totti, Maria Laura Angelini, Erica Brugin, Patrizio Sarto, Laura Merlo, Sergio Sgarzi, Michele Stancari, Paola Todeschini, Gaetano La Manna, Andrea Ermolao, Ferdinando Tripi, Lucia Andreoli, Gianluigi Sella, Alberto Anedda, Laura Stefani, Giorgio Galanti, Rocco Di Michele, Franco Merni, Manuela Trerotola, Daniela Storani, Alessandro Nanni Costa World J Transplant. 2018 Feb 24; 8(1): 13–22. Published online 2018 Feb 24. doi: 10.5500/wjt.v8.i1.13  

Author Response

Reviewer #1:

The manuscript is interesting  in the  fields  of  the  exercise prescription application,  in Sports  Medicine  and also  in the cardiological  and internal  medicine . It can be improved with the addition of more details of the potential application in the metabolic chronic diseases where also the sympatho-vagal balance can be affected: as in case of diabetes, metabolic syndrome and obesity.

Thank you for your helpful comments to improve the manuscript.

In addition, it could be helpful to give some information about the impact of the model to the VO2 max parameter, if available in literature.

I searched the literature regarding the impact of diaphragmatic breathing (DB) on VO2max. Although there were no eligible studies which investigate the single effect of DB on VO2max; however, two studies examined the effect of respiratory muscle trainings on VO2max in healthy individuals. I have added the following text to 3.2.4. Healthy individuals part:

“Previous studies have shown that DB has no significant impacts on aerobic capacity in healthy individuals. Respiratory muscle trainings also appear to have no beneficial effects on VO2max in healthy non-smokers [103] and athletes [104].”

 The same aspect could be highlighted in a session dedicated to the congenital cardiopaties, especially in the post-surgical treatment.

I could not find any eligible studies investigating the effect of DB on post-operative functions in patients with congenital heart disease. However, Ait Ali et al. [41] showed that DB improved cardiorespiratory performance in post-Fontan patients. I have added the following text to the Introduction section:

“… and cardiorespiratory performance in post-Fontan patients [41].”

Also in the potential application in the post-surgical solid organ transplantation management by the physical activity program.

Unfortunately, no studies have investigated the effect of breathing exercises including DB in patients who received an organ transplant. There are some clinical trials assessing the effect of pulmonary rehabilitation on quality of life or lung function; however, that is beyond the scope of this paper.

This last aspect has been only partially evaluated in a previous study that you can insert in the text just to start with the discussion.

Renal function and physical fitness after 12-mo supervised training in kidney transplant recipients Giulio Sergio Roi, Giovanni Mosconi, Valentina Totti, Maria Laura Angelini, Erica Brugin, Patrizio Sarto, Laura Merlo, Sergio Sgarzi, Michele Stancari, Paola Todeschini, Gaetano La Manna, Andrea Ermolao, Ferdinando Tripi, Lucia Andreoli, Gianluigi Sella, Alberto Anedda, Laura Stefani, Giorgio Galanti, Rocco Di Michele, Franco Merni, Manuela Trerotola, Daniela Storani, Alessandro Nanni Costa World J Transplant. 2018 Feb 24; 8(1): 13–22. Published online 2018 Feb 24. doi: 10.5500/wjt.v8.i1.13   .

According to your comment, I have added the following sentences to the Discussion section:

“Despite these limitations, DB has the potential to improve various kinds of disease. Recently, a number of studies have shown that physical rehabilitation improves exercise capacity in transplant recipients and candidates [116,117]. DB could also be safe and feasible in the post-transplant management due to its non-invasive technique.”

Reviewer 2 Report

I think it is an interesting job for health professionals. I congratulate the author for the work done.

Author Response

Reviewer #2:

I think it is an interesting job for health professionals. I congratulate the author for the work done.

Thank you for taking the time and effort to carefully read and review my manuscript.

Reviewer 3 Report

Summary: This review report aims to offer a systematic review about contribution of diaphragmatic breathing on health.

General comments: It is a first approach for systematic reviews and more recent studies not yet included. However, it requires much more work in the refinement of the method used, and it is much advisable a greater reduction of the amplitude of the focus of the systematic review

Specific comments:
Lines 39-41- diaphragmatic breathing not sufficiently explained
Lines 52-53- as a martial arts practitioner, the author should notice that there is an enormous variety of breathing techniques that involves diaphragmatic muscle, however, with different frequencies and relative rhythms. Figure 1 denotes that, its “visual abstract” and respective legend should be ameliorated, at least be in accordance with previous (although incomplete) text about the diaphragmatic breathing.
Line 54- health is a very ample concept, this review flows along some and very different health topics like respiratory capacity and quality of life, which bring difficulties to the focus in a strong analysis.
Lines 61-67- clearly identify systematic review method; exclusion and inclusion criteria (particularly, the statistical ones); sources used and date interval of systematic reviews and new RCTs; exclusion and inclusion criteria (particularly, the statistical ones) of the systematic reviews included.
Lines 103-105- these informations are important because systematic reviews included have different criteria of inclusion, and some problems are relevant, e.g., Prem et al. did not performed meta-analysis, and the included study of Thomas et al. summed diaphragmatic breathing with nasal breathing techniques.
Line 113- notice also systematic review 47 “None of these studies evaluated the single effect of DB.”
Lines 135-136- why dating is important? Because methods evolve and quality of data analysis also. Studies from fifties and sixties may have important information but more recent studies offer enhanced methods (e.g., the non-linear ones) and statistical techniques. What is happening in this review article is that different methods for different systematic reviews are mixed, preventing discernment. One of the purposes of a systematic review is to really select and analyse eligible and comparable data.
Lines 465-466- diaphragmatic breathing is a technique; till now, a non-clinical practice; being part of non-traditional medicine.

Author Response

Reviewer #3:

Summary: This review report aims to offer a systematic review about contribution of diaphragmatic breathing on health.

General comments: It is a first approach for systematic reviews and more recent studies not yet included. However, it requires much more work in the refinement of the method used, and it is much advisable a greater reduction of the amplitude of the focus of the systematic review.

Thank you very much for taking the time and effort to carefully read and review my manuscript.

With all due respect, as I have stated in the original version of the manuscript, this review is not a systematic review but a narrative one. I did not conduct a systematic search; thus, there are some disadvantages such as selection bias and lack of rigorous standards of objectivity. However, in accordance with your comments, I have changed the title to “Effects of Diaphragmatic Breathing on Health: A Narrative Review” and have made revisions to the manuscript.

Specific comments:

Lines 39-41- diaphragmatic breathing not sufficiently explained

According to your comment, I have added the following sentence to the explanation:

“During breathing, practitioners should be careful that chest remains as still as possible and stomach moves against the hand focusing on contracting the diaphragm.”

Lines 52-53- as a martial arts practitioner, the author should notice that there is an enormous variety of breathing techniques that involves diaphragmatic muscle, however, with different frequencies and relative rhythms. Figure 1 denotes that, its “visual abstract” and respective legend should be ameliorated, at least be in accordance with previous (although incomplete) text about the diaphragmatic breathing.

According to your comment, I have replaced the picture and revised the Figure legend.

Line 54- health is a very ample concept, this review flows along some and very different health topics like respiratory capacity and quality of life, which bring difficulties to the focus in a strong analysis.

I have revised the text as follows:

“This review aims to summarize the current evidence of the impact of DB on diseases as described above as well as respiratory function and to discuss its future perspective.”

Lines 61-67- clearly identify systematic review method; exclusion and inclusion criteria (particularly, the statistical ones); sources used and date interval of systematic reviews and new RCTs; exclusion and inclusion criteria (particularly, the statistical ones) of the systematic reviews included.

As mentioned above, this paper is not a systematic review. I described the search strategy for this narrative review in the Methods section. According to your comment, I have added information regarding the sources used and date of search to the Methods section.

Lines 103-105- these informations are important because systematic reviews included have different criteria of inclusion, and some problems are relevant, e.g., Prem et al. did not performed meta-analysis, and the included study of Thomas et al. summed diaphragmatic breathing with nasal breathing techniques.

Line 113- notice also systematic review 47 “None of these studies evaluated the single effect of DB.”

Lines 135-136- why dating is important? Because methods evolve and quality of data analysis also. Studies from fifties and sixties may have important information but more recent studies offer enhanced methods (e.g., the non-linear ones) and statistical techniques. What is happening in this review article is that different methods for different systematic reviews are mixed, preventing discernment. One of the purposes of a systematic review is to really select and analyse eligible and comparable data.

Thank you for your important comments. I have added your indications to the Discussion section. In my opinion, it is also important to inform readers that there are problems in previous systematic reviews to clarify the effect of DB on various diseases.

“…Also, systematic reviews included have different criteria of inclusion. For example, several studies that did not investigate the single effect of DB were included in a systematic review (e.g. Thomas et al. [31]). Fourthly, systematic reviews included a wide range of studies from the 1950s to 2010s. Studies that were performed 50 years ago have important information; however, recent studies may be more important because statistical methods and quality of data advances with the times. Such heterogeneity might cause the findings of this review to be inconclusive.”

Lines 465-466- diaphragmatic breathing is a technique; till now, a non-clinical practice; being part of non-traditional medicine.

According to your comment, I have changed the wording as follows:

“Furthermore, DB could be a feasible and practical technique for patients with such disorders. Although further studies are needed to clarify the effects of DB on human health, DB can support clinical practice.”

Reviewer 4 Report

Dear author,

As it is often the case with meta-reviews compiling earlier researches, the heterogeneity of the results needs to be balanced with hindsight and expertise from your part. Not being an expert myself, I know that breathing techniques have been used for centuries as therapeutic tools. It would be useful to add some historical context in the introduction, in order to signal that you only focus on medically-monitored and -controlled use of DB. DB is used in yoga courses, by psychotherapists, as a way to find sleep or simply to relax. Its benefits have thus been known for centuries, the main goal of your article will therefore be to measure the full extent of these benefits and if DB can be used as a "medicine", according to the meaning evidence-based medicine gives to that word.

Besides, I'm not very sure about the structure. You provide the reader with a catalogue of contexts in which DB has been experimented under the supervision of health researchers. However, there are significant differences between the protocols, the cohorts under study and the methods used to evaluate the efficacy of DB. Your role is to find the common points between all these wide-ranging studies, and to tell the reader what facts about DB are medically valid. Even saying "well, we don't know" is a fact.

Considering what I just wrote, you will understand why I find your conclusion a bit underwhelming. It should be enriched. According to me, the "discussion" slot is the necessary space for that.

All these remarks notwithstanding, your article is notable for its wealth of information, its thoroughness, its clarity and the quality of expression. However, in my modest opinion, it needs a reshuffle.

Line 99: there is a percentage sign at the end of FEV1. Is it on purpose or is it a mistake?

Regards,

Author Response

Reviewer #4:

As it is often the case with meta-reviews compiling earlier researches, the heterogeneity of the results needs to be balanced with hindsight and expertise from your part. Not being an expert myself, I know that breathing techniques have been used for centuries as therapeutic tools. It would be useful to add some historical context in the introduction, in order to signal that you only focus on medically-monitored and -controlled use of DB. DB is used in yoga courses, by psychotherapists, as a way to find sleep or simply to relax. Its benefits have thus been known for centuries, the main goal of your article will therefore be to measure the full extent of these benefits and if DB can be used as a "medicine", according to the meaning evidence-based medicine gives to that word.

I thank you for your comments, which have helped me to improve the manuscript.

According to your comment, I have added some historical context in the introduction as follows:

“In the end of 19th century, Sewall and Pollard [1] firstly investigated the relationship between the movement of diaphragm and chest during respiration… Since time out of mind, traditional martial arts such as tai chi and yoga utilize DB in their practice.”

Besides, I'm not very sure about the structure. You provide the reader with a catalogue of contexts in which DB has been experimented under the supervision of health researchers. However, there are significant differences between the protocols, the cohorts under study and the methods used to evaluate the efficacy of DB. Your role is to find the common points between all these wide-ranging studies, and to tell the reader what facts about DB are medically valid. Even saying "well, we don't know" is a fact.

Considering what I just wrote, you will understand why I find your conclusion a bit underwhelming. It should be enriched. According to me, the "discussion" slot is the necessary space for that.

According to your comment, I have revised the study limitation part in the Discussion section. As you pointed out, the findings of this review might be inconclusive due to the heterogeneity of previous studies; however, in my opinion, it is important to inform readers that there are some problems in previous systematic reviews and clinical trials.

All these remarks notwithstanding, your article is notable for its wealth of information, its thoroughness, its clarity and the quality of expression. However, in my modest opinion, it needs a reshuffle.

I really appreciate your helpful comments. I would appreciate it if you would check the revised manuscript.

Line 99: there is a percentage sign at the end of FEV1. Is it on purpose or is it a mistake?

Thank you for your careful reviewing. FEV1.0% indicates forced expiratory volume% in 1 second. It is on purpose.

Round 2

Reviewer 3 Report

The document will allow the author to start to understand the non-clinical potential of breathing techniques based on diaphragmatic control, however, the author does not have sufficient knowlegde about research methods to transform it in an adequate review study.

Author Response

Thank you for your comment on the revised manuscript. I give my utmost thanks for your dedication to improve my manuscript.

However, I am afraid that your suggestions do not apply to my manuscript. This review is a narrative one. I did not plan to write a systematic review paper when receiving the invitation to write a narrative review paper from Medicines.

According to recent papers, the literature search methods should be included in narrative reviews as well as systematic review. Description of the search strategy is useful for reducing selection bias and defining clinical question; therefore, a structured approach on the lines of that used for systematic reviews is advisable in literature search for narrative reviews (1-4).

References

  1. Gasparyan AY, Ayvazyan L, Blackmore H, Kitas GD. Writing a narrative biomedical review: considerations for authors, peer reviewers, and editors. Rheumatol Int 2011;31:1409–1417.
  2. Liumbruno GM, Velati C, Pasqualetti P, Franchini M. How to write a scientific manuscript for publication. Blood Trans 2013;11:217–226.
  3. Murphy CM. Writing an effective review article. J Med Toxicol 2012;8:89–90.
  4. Ferrari R. Writing narrative style literature reviews. Medical Writing 2015;24:230-235.